# Design and Synthesis of Benzimidazole-Chalcone Derivatives as Potential Anticancer Agents

**DOI:** 10.3390/molecules24183259

**Published:** 2019-09-06

**Authors:** Cheng-Ying Hsieh, Pi-Wen Ko, Yu-Jui Chang, Mohit Kapoor, Yu-Chuan Liang, Hsueh-Liang Chu, Hui-Hsien Lin, Jia-Cherng Horng, Ming-Hua Hsu

**Affiliations:** 1Department of Chemistry, National Tsing Hua University, Hsinchu 30013, Taiwan; futariwhisper@gmail.com (C.-Y.H.); rick5569268@gmail.com (Y.-J.C.); 2Department of Biomedical Engineering and Environmental Sciences, National Tsing Hua University, Hsinchu 30013, Taiwan; koko37bebe@hotmail.com; 3Chitkara University Institute of Engineering and Technology, Chitkara University, Punjab 140 401, India; mohitkapoor.chemistry@gmail.com; 4Agricultural Biotechnology Research Center, Academia Sinica, Taipei 11529, Taiwan; ycliang@sinica.edu.tw; 5Graduate Institute of Translational Medicine, College of Medicine and Technology, Taipei Medical University, Taipei 11031, Taiwan; szxchu@gmail.com; 6Division of Radiotherapy, Department of Oncology, Taipei Veterans General Hospital, Taipei 11217, Taiwan; twwarcgogo@gmail.com; 7Department of Chemistry, National Changhua University of Education, Changhua 50007, Taiwan

**Keywords:** anticancer activity, benzimidazole, chalcone, human lung carcinoma, human breast adenocarcinoma, human liver carcinoma, human ovarian carcinoma

## Abstract

Numerous reports have shown that conjugated benzimidazole derivatives possess various kinds of biological activities, including anticancer properties. In this report, we designed and synthesized 24 new molecules comprising a benzimidazole ring, arene, and alkyl chain-bearing cyclic moieties. The results showed that the *N*-substituted benzimidazole derivatives bearing an alkyl chain and a nitrogen-containing 5- or 6-membered ring enhanced the cytotoxic effects on human breast adenocarcinoma (MCF-7) and human ovarian carcinoma (OVCAR-3) cell lines. Among the 24 synthesized compounds, (2*E*)-1-(1-(3-morpholinopropyl)-1*H*-benzimidazol-2 -yl)-3-phenyl-2-propen-1-one) (**23a**) reduced the proliferation of MCF-7 and OVCAR-3 cell lines demonstrating superior outcomes to those of cisplatin.

## 1. Introduction

Cancer is the uncontrolled growth of abnormal cells leading to profound changes in physiological function. Cancer cells, which can evade the immune system, influence the normal cells, molecules, and blood vessels that surround and feed a tumor. The tumors, which can grow and metastasize to other locations in the body, can potentially interfere with the digestive, nervous, and circulatory systems and release hormones that alter body functions. For anticancer drug development, designing molecules that can selectively inhibit the proliferation of abnormal cells with minimal or no effect on normal cells is critical. Therefore, developing anticancer drugs is of utmost importance worldwide [1,2,3].

Benzimidazole ring systems possess various applications in novel drug development. Benzimidazole is a naturally occurring bicyclic compound [4] consisting of a fused benzene and imidazole ring and is an integral part of vitamin B_12_. Because of the structural similarities of benzimidazoles with purine, they can easily interact with the biomolecules of living systems. Therefore, they have considerable potential for use in medicinal chemistry and are a critical pharmacophore in drug discovery [5,6,7].

Benzimidazole and its derivatives possess various biological activities, including antibacterial, [8] anti-tubercular, [9,10] antifungal, [11] antimicrobial, [9,11] antiprotozoal, [12,13] anti-HIV, [14] and antiviral activities, [15] and show potential as protein kinase inhibitors. [16] Achar et al. synthesized several 2-methylaminobenzimidazole derivatives (Figure 1, **1**), that displayed potent in-vivo analgesic and anti-inflammatory activities. [17] On the basis of this information, Refaat et al. designed benzylidene cyanomethylbenzimidazole (Figure 1, **2**), which showed excellent anticancer activity potential against human liver carcinoma (HEP-G2) cell lines. [18] Azam et al. reported that phenylbenzimidazole analogues (Figure 1, **3**) possess potent anti-cancer activity against five human cancer cell lines. [19] Various potent drugs containing a benzimidazole nucleus are currently available on the market. For example, albendazole (for treatment of certain infections caused by worms such as pork tapeworm and dog tapeworm, Figure 1, **4**) [13], omeprazole (a member of the group of drugs called proton pump inhibitors that decrease the amount of acid produced in the stomach, Figure 1, **5**), and pimobendan (a calcium sensitizer and selective inhibitor of phosphodiesterase III with positive inotropic and vasodilator effects, Figure 1, **6**). Albendazole also inhibits hepatocellular carcinoma cell proliferation under both in vitro and in vivo experimental conditions. [20] Therefore, the optimization of benzimidazole derivatives on the basis of their structures has attracted considerable attention in recent years.

Chalcones (Figure 2, **7**) which belong to the flavonoid family, are open-chained molecules bearing an α,β-unsaturated carbonyl system between two aromatic rings and represent a crucial class of molecules that are abundant in edible plants. [21] Chalcones reportedly possess beneficial biological activities [22], particularly considering their similar mode of action to the structurally related natural phenol combretastatin (Figure 2, **8**). [23] Many studies on chalcone derivatives have shown that they demonstrate antibacterial [24], antimalarial [25,26], antifungal [27], anti-HIV [28], anti-inflammatory [29], and anticancer activities [30,31,32,33]. Modzelewska et al. synthesized a series of bis-chalcones (Figure, **9**) that demonstrated exceptional performance in inhibiting of the growth of human breast and colon cancer cells. [32]

On the basis of a literature survey, we synthesized benzimidazole-chalcone conjugates, which could lead to the development of potent anti-cancer agents. In the development of new drugs, combining various moieties with different biological activities may lead to the creation of novel candidates with exceptional pharmacological activity [34,35,36,37]. Some recent studies have proposed several hybrid benzimidazolyl-chalcone derivatives that display anthelmintic [38], antifungal [39] and antitumor activities [40]. Woo et al. showed that substituted benzimidazolyl curcumin mimics (Figure 3, **10**) possess anticancer activity, and they hypothesized that the increment in inhibitory potency is due to the attached benzimidazole functionalities [41]. In addition, some benzimidazole derivatives serve as antagonists of the chemokine receptor CXCR3 [42] (Figure 3, **11**), inhibitors of the hepatitis B virus [15] (Figure 3, **12**), and inhibitors of *Francisella tularensis* enoyl-ACP reductase [43] (Figure 3, **13**), and antitumor activities [44] (Figure 3, **14**) by modifying some monomers on benzimidazole. Herein, we report the scope of activity of benzimidazole-chalcone conjugates as anti-cancer agents. Through an analysis of their anti-cancer activities on various cell lines influenced by the substituents on phenyl and nitrogen, their structure-activity relationship (SAR) guidelines were deduced.

## 2. Results and Discussion

### 2.1. Chemistry

The synthetic strategies adopted to obtain the target compounds are depicted in Scheme 1. Syntheses of benzimidazole-chalcone derivatives were achieved through conjugation of benzimidazole and aromatic aldehydes under basic conditions. Benzimidazole derivative **16** was synthesized in high yield [45] through a reaction of *o*-phenylenediamine with lactic acid (1.1 equiv) in hydrochloric acid (4.0 N concentration) under reflux for 6.0 h, which underwent oxidation in the presence of potassium permanganate (2.5 equiv) and solid aluminium oxide to afford the intermediate compound **17**, which was prepared from the oxidation products. After being ground with a pestle and mortar and then purified, compound **17** was obtained in 72% yield as a white solid [46]. We prepared the benzimidazolyl-chalcones **18a**–**18d** in 82–95% overall yields by reacting compound **17** with substituted aromatic aldehydes possessing 4-methyl- (**18b**), 4-methoxy- (**18c**), and 4-chloro- (**18d**) groups in the presence of a base, through conventional Claisen-Schmidt condensations. [35] To expand the structural diversity of the benzimidazolyl-chalcone derivatives, we synthesized substituted benzimidazolyl-chalcones **19**–**23** from allyl bromide or various alkyl chlorides bearing a hydrocarbon spacer and a nitrogen-containing 5- or 6-membered ring (1.2 equiv) in the presence of sodium carbonate and acetonitrile. For the preparation of the final compounds, compounds **18a**–**18d** were reacted with allyl bromide, 4-(2-chloroethyl)morpholine, 1-(2-chloroethyl)-piperidine, 1-(2-chloroethyl)pyrrolidine or 4-(3-chloropropyl)morpholine (1.2 equiv) in acetonitrile with potassium carbonate (2.9 equiv) at reflux overnight to afford (2E)-1-(1-allyl-1H-benzimidazol-2-yl)-3-phenyl-2-propen-1-ones **19a**–**23a**, (2E)-1-(1-allyl-1H-benzimidazol-2-yl)-3-(4-methylphenyl)-2-propen-1-ones **19b**–**23b**, (2E)-1-(1-allyl-1H-benzimidazol-2-yl)-3-(4-methoxyphenyl)-2 -propen-1-ones **19c**–**23c**, and (2*E*)-1-(1-allyl-1*H*-benzimidazol-2-yl)-3-(4-chlorophenyl)-2-propen-1-ones **19d**–**23d**, respectively, in 50–86% overall yields. The characterization data of compounds are showed in the supporting information.

### 2.2. Pharmacology

To determine the SAR, we designed various types of benzimidazole-chalcone derivatives **18**–**23** having a phenyl ring substituted with electron-donating and electron-withdrawing groups as well as modified benzimidazolyl groups (allyl, nitrogen-containing 5- or 6-membered rings). All the new compounds were evaluated against four common types of human cancer, namely human lung carcinoma (A549), human breast adenocarcinoma (MCF-7), human hepatoma (HEP-G2), and human ovarian carcinoma (OVCAR-3) cell lines. For comparison purposes, the cytotoxicity of doxorubicin (a standard antitumor drug) and cisplatin (a platinum-containing anti-cancer drug) were evaluated under identical conditions. MTT (3-(4,5-dimethylthiazol-2-yl)-2,5-diphenyl tetrazolium bromide) assays were conducted. The MTT assay is used calculate the relationship between cell viability under different treatments [47,48,49]. The IC_50_ value (the dose of the compound that causes a 50% reduction in the survival value) found via MTT assay is used to evaluate the potential anticancer activity of compounds [50,51]. The IC_50_ values of compounds **18a**–**23d** are summarized in Table 1.

Most of the compounds demonstrated potential antitumor activities against all of the tested tumor cell lines. Among the 24 benzimidazole derivatives, eight compounds showed IC_50_ values in the 9.73–12.47 μM range and six compounds in the 14.59–19.53 μM range on the A549 cell line. The IC_50_ of these 24 compounds on the MCF-7 cell line demonstrated promising activities; 10 of the compounds showed an IC_50_ between 8.91 and 12.12 μM. Regarding the HEP-G2 cell line, three compounds showed an IC_50_ in the 10.16–10.93 μM range. Finally, the IC_50_ values of these 24 compounds for the OVCAR-3 cell line were as follows: 11 compounds showed an IC_50_ in the range of 10.34–14.88 μM, 10 in the range of 16.09–36.48 μM, and the remaining compounds were higher than 42.24 μM. The differences in the IC_50_ values may be attributable to such factors as the nature of the N-substitution at the benzimidazole ring system and the functionality of the phenyl moiety on the 4 position of the phenyl ring system, and the genetic and biochemical background of the cell lines.

Generally, most of the tested compounds tended to be more active against MCF-7 and OVCAR-3 than against the other tumor cell lines. Six compounds (**20a**–**23a, 22b, 22c**) showed IC_50_ values of less than 11.70 μM (the IC_50_ value of cisplatin) on the MCF-7 cell line. In contrast to the OVCAR-3 cell line, 11 compounds (**20a**, **21a**, **23a**, **21b**, **23b**, **20c**–**23c, 20d**, **22d**) had IC_50_ values between 10.34 and 14.88 μM, which were less than 16.04 μM (the IC_50_ value of cisplatin). The most active compound among the tested benzimidazole derivatives was **23a**, which exhibited IC_50_ values of 9.73, 8.91, 10.93 and 10.76 μM on the A549, MCF-7, HEP-G2 and OVCAR-3 cells, respectively. This compound showed in vitro cytotoxicity comparable or superior to that of cisplatin.

SAR studies have suggested that the introduction of a nitrogen-containing 5- or 6-membered ring bearing a hydrocarbon spacer group (**20a, 21a, 23a**) in the N-substitution pattern of (2E)-1-(1H-benzimidazol-2-yl)-3-phenyl-2-propen-1-one (**18a**) leads to an increase in cytotoxic activity on A549, MCF-7, and OVCAR-3 cells, compared with the unsubstituted compound **18a**. Compared with compounds **23a**–**23d**, (2E)-1-(1-(3-morpholinopropyl)-1H-benzimidazol-2-yl)-3-phenyl-2-propen-1-one (**23a**) was generally more active than the 4-methyl-, 4-methoxy- and 4-chloro- functionality on phenyl moieties (compounds **23b**–**23d**) in A549, MCF-7 and OVCAR-3 cells. Similarly, among compounds **21a**–**21d** and **20a**–**20d**, we observed the same trend as in compounds **23a**–**23d**. This result suggests that the hybridization of the chalcone and benzimidazole ring system, which is modified with a biological side chain bearing a hydrocarbon spacer and a nitrogen-containing 5- or 6-membered ring in the N-substitution, can lead to enhanced cytotoxic effects on MCF-7 and OVCAR-3 cells. Notably, these data show that some of the tested compounds are more potent than cisplatin.

In pharmacotherapy, most drugs are taken via oral administration and are absorbed through the gastrointestinal (GI) system [52]. Some drugs have to cross a series of barriers either by passive diffusion or carrier conjugation [53]. Therefore, lipophilicity is a very important index in anticancer drug development [52,53,54]. The ALOGPS software package predicts lipophilicity and aqueous solubility of chemical compounds [53,54,55]. In this software, logP, is accepted as the principal parameter through which to evaluate lipophilicity of chemical compounds which largely determines the pharmacokinetic properties of drugs [53].

In this study, no significant correlation was observed between the IC_50_ values and log P values of the 24 compounds (Table 1). Therefore, the difference in lipophilicity might not be a significant factor in the differences in cytotoxicity of the tested compounds. By contrast, we observed no correlation between the IC_50_ and water solubility values of these compounds, which differs from the modified benzimidazole ring system. However, in these compounds, which differ in the 4-methyl-, 4-methoxy-, and 4-chloro- functionality of their phenyl moiety, we observed a correlation between the IC_50_ and water solubility values (S (cal), Table 1). For example, the IC_50_ values of **23a** (8.03 mmol/L water solubility) and **23b**–**23d** (3.26–6.00 mmol/L water solubility) were 8.91 versus 11.34–35.69 μM (MCF-7) and 10.76 versus 13.76–42.24 μM (OVCAR-3), respectively (Table 1). Similarly, among compounds **18a**–**18d**, **20a**–**20d** and **20c, 21a, 21b, 21d**, we observed the same trend as compared with compounds **23a**–**23d**. Therefore, the difference in water solubility may be a factor explaining the difference in cytotoxicity of the benzimidazole derivatives examined in our study.

We conducted a cell cycle analysis through flow cytometry of compounds **20a**–**23a** to elucidate their mechanism of action. OVCAR-3 cells were treated with compounds **20a**–**23a** for 48 h. The histograms obtained from the OVCAR-3 cells demonstrated a major peak (G_1_) and minor peak (G_2_) (Appendix A). OVCAR-3 cells treated with 10.50 μM compound **20a** and 10.34 μM compound **21a** showed 55% and 63% in the G2/M phase, respectively, whereas an untreated control showed 37% in the G2/M phase with same value of 8% in the S phase (Table 2). By contrast, treatment of OVCAR-3 cells with the 22.44 μM compound **22a** and 10.76 μM compound **23a** did not alter the cell cycle distribution compared with the untreated control. Because compounds **20a** and **21a** could effectively induce cell cycle arrest and inhibit cancer cell growth at a similar concentration, our data suggest that the growth inhibition activities of compounds **20a** and **21a** are directly related to their abilities to arrest cell cycle progression. In contrast to **20a** and **21a**, compound **23a** did not induce cell cycle arrest at 10 μM on OVCAR-3, whereas the IC_50_ values of compound **23a** for cancer cells were observed at 10.76 μM (Table 1), therefore, the cytotoxic effects of **23a** might not be due to cell cycle arrest.

## 3. Materials and Methods

### 3.1. General Procedures

All reactions were carried out in oven-dried glassware (120 °C) under an atmosphere of nitrogen. Acetone, acetonitrile, ethanol, dichloromethane, ethyl acetate and hexane from Mallinckrodt Chemical Co. (Staines-Upon-Thames, United Kingdom) were dried and distilled from CaH_2_. Allyl bromide, benzaldehyde, 4-chlorobenzaldehyde, 4-(2-chloroethyl)morpholine, 4-(3-chloropropyl)morpholine, 1-(2-chloroethyl)-piperidine, 1-(2-chloroethyl)pyrrolidine, lactic acid, 4-methylbenzaldehyde, 4-methoxybenzaldehyde,*o*-phenylenediamine, potassium carbonate, and potassium permanganate were purchased from Sigma-Aldrich Chemical Co. (St. Louis, MO, USA). Hydrochloric acid, magnesium sulfate, potassium hydroxide, and sodium hydroxide were purchased from Showa (Tokyo, Japan). Aluminum oxide was purchased from Merck (Darmstadt, Germany).

Analytical thin layer chromatography (TLC) was performed on precoated plates (silica gel 60 F-254), purchased from Merck. Purification by gravity column chromatography was carried out by use of Merck Silica Gel 60 (particle size 0.063–0.200 mm, 70–230 mesh ASTM). ^1^H-NMR spectra were obtained on an Avance 500 (500 MHz) spectrometer (Bruker, Billerica, MA, USA) by use of chloroform-d all examples given use acetone and acetone-*d_6_* as solvents. ^1^H-NMR chemical shifts were referenced to the CDCl_3_ singlet (7.24 ppm) and the acetone-*d_6_* quintet (2.05 ppm). ^13^C-NMR spectra were obtained on Bruker Avance 500 (125 MHz), AM-400* (Bruker) and MR-400 (Varian, Palo Alto, CA, USA) all examples given use a 500 MHz instrument spectrometers by use of acetone-*d_6_* as solvent. ^13^C-NMR chemical shifts were referenced to the center of the acetone-*d_6_* septet (29.92 ppm). Multiplicities were recorded by the following abbreviations: brs, broad singlet; s, singlet; d, doublet; t, triplet; q, quartet; m, multiplet; *J*, coupling constant (hertz). High-resolution mass spectra were obtained by means of a FINNIGAN/MAT-95XL mass spectrometer (Thermo Fisher Scientific, Waltham, MA, USA). High-performance liquid chromatography (HPLC) analyses were carried out on an 1100 series system (Agilent, Santa Clara, CA, USA) equipped with a CNW Athena C18 column (120 Å, 4.6 mm × 250 mm, 5 μm) and UV detection at 254 nm. A mixture of 20% DI water in acetonitrile was used as eluent and flow rate was at 0.5 mL/min. Infrared (IR) spectra were measured on a RX 1FT-IR spectrometer (Perkin Elmer, Waltham, MA, USA). Absorption intensities were recorded using the following abbreviations: s, strong; m, medium; w, weak; br, broad.

### 3.2. Synthesis

#### 3.2.1. Synthesis of 1-(1*H*-benzoimidazol-2-yl)ethan-1-ol (**16**).

Hydrochloric acid (4.0 N, 25 mL) was added to a stirred solution of *o*-phenylenediamine (4.32 g, 40.0 mmol, 1.0 equiv) in lactic acid (3.96 g, 44.0 mmol, 1.1 equiv) and the reaction mixture was heated to reflux for 16 h. After 16 h, the reaction mixture was cooled down to room temperature and neutralized with sodium hydroxide solution. The reaction mass was filtered to obtain compound **16** (6.15 g, 38.0 mmol) in 95% yield as a pale yellow solid: ^1^H-NMR (CDCl_3_, 500 MHz) δ 1.73 (d, *J* = 6.0 Hz, 3 H, CH_3_), 5.22 (q, *J* = 6.0 Hz, 1 H, CH), 7.26 (d, *J* = 6.0 Hz, 1 H, ArCH), 7.27 (d, *J* = 5.5 Hz, 1 H, ArCH), 7.59 (d, *J* = 5.5 Hz, 1 H, ArCH), 7.60 (d, *J* = 6.0 Hz, 1 H, ArCH).

#### 3.2.2. Synthesis of 1-(1*H*-benzimidazol-2-yl)ethan-1-one (**17**).

First, alumina-supported permanganate was prepared by mixing solid KMnO_4_ (2.0 g, 12.65 mmol, 2.5 equiv) and solid aluminium oxide (2.5 g) in a mortar ground with a pestle for 3.0 min. Then compound (**16**, 0.810 g, 4.99 mmol, 1.0 equiv) was added in the mortar and stirred for another 10 min. Acetone (40 mL) was added to the reaction mixture in a beaker and stirred for 20 min. The mixture was filtered and the filtrate was evaporated to obtain a crude residue. The organic mass was extracted with EtOAc (2 × 10 mL) and washed with H_2_O (2 × 5.0 mL), dried over MgSO_4(s)_, filtered, and concentrated under reduced pressure. The residue was purified by use of column chromatography (10% ethyl acetate in hexane as eluent) to give the desired compound **17** (0.580 g, 3.62 mmol) in 72% yield as white solids. ^1^H-NMR (CDCl_3_, 400 MHz) δ 2.81 (s, 3 H, CH_3_), 7.35 (dd, *J* = 8.0 Hz, 1 H, ArCH), 7.41 (dd, *J* = 7.5 Hz, 1 H, ArCH), 7.53 (d, *J* = 8.0 Hz, 1 H, ArCH), 7.90 (d, *J* = 8.0 Hz, 1 H, ArCH).

#### 3.2.3. Standard Procedure 1 for the Synthesis of Benzimidazolyl-Chalcone Derivatives **18a**–**18d.**

Compound **18a–18d** had been reported in previous literatures [56]. Aqueous KOH (40%) was added to a stirred solution of 2-acetylbenzimidazole (**17**, 1.0 equiv) and substituted aromatic aldehydes (1.1 equiv) in ethanol. The resulting mixture was stirred at room temperature for 10 h. After the consumption of starting materials, the reaction mixture was quenched with water (2–4 mL) and extracted with EtOAc (10–20 mL). The combined organic layers were washed with brine (5.0 mL), dried over MgSO_4(s)_, filtered, and concentrated under reduced pressure. The residue was purified by column chromatography to give the desired benzimidazolyl-chalcone derivatives.

#### 3.2.4. Standard Procedure 2 for Synthesis of Side Chain Modified Benzimidazolyl-chalcone Derivatives **19**–**23**.

Potassium carbonate (3.0 equiv) was added to a stirred solution of benzimidazolyl-chalcone derivatives, **18a**–**18d** (1.0 equiv) and substituted halides (1.2 equiv) in acetonitrile and the resulting solution was heated at reflux overnight. The reaction mixture was cooled to room temperature, quenched with water (2.0–4.0 mL), and extracted with EtOAc (10–20 mL). The combined organic layers were washed with brine (5.0 mL), dried over MgSO_4(s)_, filtered, and concentrated under reduced pressure. The residue was purified by column chromatography to give the desired side chain modified benzimidazolyl-chalcone derivatives.

### 3.3. Compound Data

*(**2**E)-1-(1H-**b**enzimidazol-2-yl)-3-phenyl-2-propen-1-one* (**18a**). Standard procedure **1** was followed by use of compound **17** (0.640 g, 4.0 mmol, 1.0 equiv), benzaldehyde (0.467 g, 4.4 mmol, 1.1 equiv), and aq. KOH (40%, 2.0 mL) in ethanol (8.0 mL). After workup and purification with column chromatography (15% EtOAc in hexane as eluent), compound **18a** (0.870 g, 3.51 mmol) was obtained in 88% yield as yellow solids: ^1^H-NMR (acetone-*d_6_*, 500 MHz) δ 7.37 (t, *J* = 7.5 Hz, 1 H, ArCH), 7.44 (t, *J* = 7.5 Hz, 1 H, ArCH), 7.52-7.53 (m, 3 H, 3 × ArCH), 7.68 (d, *J* = 8.0 Hz, 1 H, ArCH), 7.88 (d, *J* = 7.5 Hz, 2 H, 2 × ArCH), 7.88 (d, *J* = 7.5 Hz, 1 H, ArCH), 8.03 (d, *J* = 16 Hz, 1 H, COCH), 8.17 (d, *J* = 16.0 Hz, 1 H, PhCH); ^13^C-NMR (acetone-*d_6_*, 125 MHz) δ 113.64, 122.45, 122.55, 124.18, 126.82, 128.97, 129.78, 130.08, 131.77, 131.91, 135.81, 144.70, 145.23, 146.01, 150.20, 181.92; IR (neat) 3357 (s), 3247 (N-H, s), 2920 (s), 2850 (s), 1661 (C=O, s), 1632 (m), 1597 (C=N, s), 1424 (s), 1331 (C–N, s), 1215 (m), 1138 (w), 1089 (w), 971 (w), 741 (w), 721.59 (w); MS (ESI) *m/z* calcd for C_16_H_12_N_2_O: 248.0950, found: 248.0950. Its spectroscopic characteristics are consistent with those reported for the same compound [56].

*(2E)-1-(1H-benzimidazol-2-yl)-3-(4-methylphenyl)-2-propen-1-one* (**18b**). Standard procedure **1** was followed by use of compound **17** (0.640 g, 4.0 mmol, 1.0 equiv), 4-methylbenzaldehyde (0.529 g, 4.4 mmol, 1.1 equiv), and aq. KOH (40%, 2.0 mL) in ethanol (8.0 mL). After workup and purification with column chromatography (20% EtOAc in hexane as eluent), compound **18b** (0.942 g, 3.59 mmol) was obtained in 90% yield as yellow solids: ^1^H-NMR (acetone-*d_6_*, 500 MHz) δ 2.43 (s, 3 H, CH_3_), 7.34 (d, *J* = 8.0 Hz, 2 H, 2 × ArCH), 7.39 (d, *J* = 6.0 Hz, 1 H, ArCH), 7.40 (d, *J* = 7.5 Hz, 1 H, ArCH), 7.76 (d, *J* = 7.5 Hz, 4 H, 4 × ArCH), 7.99 (d, *J* = 16 Hz, 1 H, COCH), 8.13 (d, *J* = 16 Hz, 1H, PhCH); ^13^C-NMR (acetone-*d_6_*, 125 MHz) δ 21.60, 113.63, 121.56, 122.42, 124.15, 126.75, 129.86, 130.78, 132.23, 133.14, 135.80, 142.52, 144.73, 145.33, 181.92; IR (neat) 3839 (s), 3736 (s), 3649 (m), 3568 (m), 3362 (w), 3247 (N–H, br), 2920 (s), 2846 (s), 2363 (s), 1657 (C=O, s), 1594 (C=N, m), 1424 (w), 1328 (C–N, m), 1218 (s), 1089 (m), 982 (m), 809 (w), 738 (m); MS (ESI) *m/z* calcd for C_17_H_14_N_2_O: 262.1106, found: 262.1107. Its spectroscopic characteristics are consistent with those reported for the same compound [56].

*(2E)-1-(1H-benzimidazol-2-yl)-3-(4-meth**oxy**phenyl)-2-propen-1-one* (**18c**). Standard procedure **1** was followed by use of compound **17** (0.640 g, 4.0 mmol, 1.0 equiv), 4-methoxybenzaldehyde (0.598 g, 4.4 mmol, 1.1 equiv), and aq. KOH (40%, 2.0 mL) in ethanol (8.0 mL). After workup and purification with column chromatography (20% EtOAc in hexane as eluent), compound **18c** (1.05 g, 3.78 mmol) was obtained in 95% yield as yellow solids: ^1^H-NMR (acetone-*d_6_*, 500 MHz) δ 3.90 (s, 3 H, OCH_3_), 7.08 (d, *J* = 9.0 Hz, 2 H, 2 × ArCH), 7.35 (t, *J* = 7.5 Hz, 1 H, ArCH), 7.42 (t, *J* = 7.5 Hz, 1 H, ArCH), 7.68 (d, *J* = 8.5 Hz, 1 H, ArCH), 7.84 (d, *J* = 9.0 Hz, 2 H, 2 × ArCH), 7.87 (d, *J* = 8.0 Hz, 1 H, ArCH), 7.99 (d, *J* = 16 Hz, 1 H, COCH), 8.04 (d, *J* = 16 Hz, 1 H, PhCH); ^13^C-NMR (acetone-*d_6_*, 125 MHz) δ 55.97, 113.59, 115.56, 120.09, 122.37, 124.08, 126.63, 128.45, 131.72, 135.76, 144.73, 145.21, 150.44, 163.29, 181.81; IR (neat) 3357 (s), 3192 (N–H, s), 2920 (s), 2850 (s), 2313 (w), 1756 (br), 1657 (C=O, s), 1634 (s), 1586 (C=N, m), 1514 (m), 1468 (m), 1421 (m), 1369 (w) 1322 (C–N, w) 1243 (s), 1172 (m), 1089 (w), 1056 (w), 823 (w), 718 (w), 631 (w); MS (ESI) *m/z* calcd for C_17_H_14_N_2_O_2_: 278.1055, found: 278.1055. Its spectroscopic characteristics are consistent with those reported for the same compound [56].

*(2E)-1-(1H-benzimidazol-2-yl)-3-(4-**chloro**phenyl)-2-propen-1-one* (**18d**). Standard procedure **1** was followed by use of compound **17** (0.640 g, 4.0 mmol, 1.0 equiv), 4-chlorobenzaldehyde (0.616 g, 4.4 mmol, 1.1 equiv), and aq. KOH (40%, 2.0 mL) in ethanol (8.0 mL). After workup and purification with column chromatography (20% EtOAc in hexane as eluent), compound **18d** (0.976 g, 3.45 mmol) was obtained in 86% yield as yellow solids: ^1^H-NMR (acetone-*d_6_*, 500 MHz) δ 7.37 (t, *J* = 7.5 Hz, 1 H, ArCH), 7.44 (t, *J* = 7.5 Hz, 1 H, ArCH), 7.55 (d, *J* = 8.5 Hz, 2 H, 2 × ArCH), 7.68 (d, *J* = 8.5 Hz, 1 H, ArCH), 7.88 (d, *J* = 8.5 Hz, 1 H, ArCH), 7.89 (d, *J* = 8.5 Hz, 2 H, 2 × ArCH), 7.99 (d, *J* = 16 Hz, 1 H, COCH), 8.16 (d, *J* = 16 Hz, 1 H, PhCH); ^13^C-NMR (acetone-*d_6_*, 125 MHz) δ 113.55, 122.37, 123.18, 124.12, 126.79, 128.95, 130.10, 131.23, 133.33, 134.59, 135.72, 137.01, 143.54, 144.61, 149.99, 181.69; IR (neat) 3841 (s), 3736 (s), 3648 (m), 2918 (w), 2846 (w), 2357 (s), 1772 (w), 1646 (C=O, s), 1599 (C=N, s), 1501 (br), 1405 (m), 1327 (C–N, s) 1218 (s), 1084 (s), 977 (s), 823 (s) 765 (w), 744 (s) 642 (w), 546 (w); MS (ESI) *m/z* calcd for C_16_H_11_ClN_2_O: 282.0560, found: 282.0562. Its spectroscopic characteristics are consistent with those reported for the same compound [56].

*(2E)-1-(1-Allyl-1H-benzimidazol-2-yl)-3-phenyl-2-propen-1-one* (**19a**). Standard procedure **2** was followed by use of compound **18a** (0.124 g, 0.50 mmol, 1.0 equiv) and allyl bromide (0.0726 g, 0.60 mmol, 1.2 equiv) in acetonitrile (15 mL) to which potassium carbonate (0.200 g, 1.45 mmol, 2.9 equiv) was added. After workup and purification with column chromatography (20% EtOAc in hexane as eluent), compound **19a** (0.110 g, 0.38 mmol) was obtained in 76% yield as yellow solids: ^1^H-NMR (acetone-*d_6_*, 500 MHz) δ 5.07 (d, *J* = 17.0 Hz, 1 H, CHCH_2(trans)_), 5.16 (d, *J* = 10.0 Hz, 1 H, CHCH_2(cis)_), 5.44 (d, *J* = 5.0 Hz, 2 H, NCH_2_), 6.09–6.16 (m, 1H, CHCH_2_), 7.40 (t, *J* = 7.5 Hz, 1 H, ArCH), 7.47 (t, *J* = 7.5 Hz, 1 H, ArCH), 7.50-7.51 (m, 3 H, 3 × ArCH), 7.67 (d, *J* = 8.5 Hz, 1 H, ArCH), 7.84 (d, *J* = 7.5 Hz, 2 H, 2 × ArCH), 7.88 (d, *J* = 8.5 Hz, 1 H, ArCH), 7.90 (d, *J* = 16.0 Hz, 1H, CCOCH), 8.31 (d, *J* = 16.0 Hz, 1H, PhCH); ^13^C-NMR (acetone-*d_6_*, 125 MHz) δ 48.26, 112.44, 117.30, 122.55, 124.13, 124.62, 126.87, 129.72, 130.06, 131.80, 134.46, 135.82, 137.54, 142.84, 144.74, 147.66, 183.18; IR (neat) 3789 (s), 3696 (w), 3663 (w), 3574 (w), 3355 (m), 2920 (s), 2850 (s), 1666 (C=O, s), 1605 (C=N, s), 1473 (w), 1449 (s), 1407 (w), 1332 (C–N, s), 1303 (w), 1165 (m), 1051 (s), 988 (m), 743 (s), 560 (m); MS (ESI) *m/z* calcd for C_19_H_16_N_2_O: 288.1263, found: 288.1264.

*(2E)-1-(1-Allyl-1H-benzimidazol-2-yl)-3-**(4-methyl**phenyl**)**-2-propen-1-one* (**19b**). Standard procedure **2** was followed by use of compound **18b** (0.131 g, 0.50 mmol, 1.0 equiv) and allyl bromide (0.0726 g, 0.60 mmol, 1.2 equiv) in acetonitrile (15 mL) to which potassium carbonate (0.200 g, 1.45 mmol, 2.9 equiv) was added. After workup and purification with column chromatography (20% EtOAc in hexane as eluent), compound **19b** (0.088 g, 0.29 mmol) was obtained in 58% yield as yellow solids: ^1^H-NMR (acetone-*d_6_*, 500 MHz) δ 2.47 (s, 3H, CH_3_), 5.06 (d, *J* = 17.0 Hz, 1 H, CHCH_2(trans)_), 5.16 (d, *J* = 10 Hz, 1 H, CHCH_2(cis)_), 5.43 (d, *J* = 5.0 Hz, 2 H, NCH_2_), 6.08-6.15 (m, 1 H, CHCH_2_), 7.32 (d, *J* = 8.0 Hz, 2 H, 2 × ArCH), 7.39 (t, *J* = 7.5 Hz, 1H, ArCH), 7.47 (t, *J* = 7.5 Hz, 1 H, ArCH), 7.66 (d, *J* = 7.5 Hz, 1 H, ArCH), 7.72 (d, *J* = 8.5 Hz, 2 H, 2 × ArCH), 7.87 (d, *J* = 16.0 Hz, 1 H, COCH), 7.88 (d, *J* = 8.0 Hz, 1H, ArCH), 8.26 (d, *J* = 16.0 Hz, 1H, PhCH); ^13^C-NMR (acetone-*d_6_*, 125 MHz) δ 21.59, 48.27, 112.40, 117.28, 122.58, 123.20, 124.53, 126.75, 129.78, 130.74, 133.17, 134.54, 135.58, 142.35, 142.94, 144.74, 147.80, 183.23; IR (neat) 3835 (s), 3731 (s), 2917 (m), 2846 (w), 2346 (s), 1748 (s), 1657 (C=O, br), 1597 (C=N, m), 1509 (w), 1375 (w), 1339 (C–N, w), 1231 (s), 1204 (w) 1180 (w), 1163 (m), 1048 (w), 985 (w), 812 (m), 752 (w), 738 (w), 724 (w); MS (ESI) *m/z* calcd for C_20_H_18_N_2_O: 302.1419, found: 302.1417.

*(2E)-1-(1-Allyl-1H-benzimidazol-2-yl)-3-**(4-methoxy**phenyl**)**-2-propen-1-one* (**19c**). The standard procedure **2** was followed by use of compound **18c** (0.139 g, 0.50 mmol, 1.0 equiv) and allyl bromide (0.0726 g, 0.60 mmol, 1.2 equiv) in acetonitrile (15 mL) to which potassium carbonate (0.200 g, 1.45 mmol, 2.9 equiv) was added. After workup and purification with column chromatography (20% EtOAc in hexane as eluent), compound **19c** (0.116 g, 0.36 mmol) was obtained in 72% yield as yellow solids: ^1^H-NMR (acetone-*d_6_*, 500 MHz) δ 3.89 (s, 3 H, OCH_3_), 5.07 (d, *J* = 17.0 Hz, 1 H, CHCH_2(trans)_), 5.16 (d, *J* = 10.0 Hz, 1 H, CHCH_2(cis)_), 5.43 (d, *J* = 5.0 Hz, 2 H, NCH_2_), 6.08–6.15 (m, 1 H, CHCH_2_), 7.06 (d, *J* = 10 Hz, 2 H, 2 × ArCH), 7.39 (t, *J* = 7.5 Hz, 1 H, ArCH), 7.46 (t, *J* = 7.5 Hz, 1 H, ArCH), 7.66 (d, *J* = 8.0 Hz, 1 H, ArCH), 7.80 (d, *J* = 8.5 Hz, 2 H, 2 × ArCH), 7.87 (d, *J* = 8.5 Hz, 1 H, ArCH), 7.87 (d, *J* = 16.0 Hz, 1 H, COCH), 8.17 (d, *J* = 16.0 Hz, 1H, PhCH); ^13^C-NMR (acetone-*d_6_*, 125 MHz) δ 48.25, 55.95, 112.38, 115.53, 117.26, 121.78, 122.51, 124.48, 126.64, 128.49, 131.64, 134.58, 137.55, 142.92, 144.68, 147.92, 163.19, 183.18; IR (neat) 3789 (s), 3357 (s), 3187 (w), 2920 (s), 2850 (s), 1659 (C=O, s), 1591 (C=N, s), 1330 (C–N, m), 1292 (m), 1254 (s), 1169 (s), 1054 (s), 988 (m), 826 (s), 741 (w); MS (ESI) *m/z* calcd for C_20_H_18_N_2_O_2_: 318.1368, found: 318.1367.

*(2E)-1-(1-Allyl-1H-benzimidazol-2-yl)-3-**(4-chloro**phenyl**)**-2-propen-1-one* (**19d**). Standard procedure **2** was followed by use of compound **18d** (0.141 g, 0.50 mmol, 1.0 equiv) and allyl bromide (0.0726 g, 0.60 mmol, 1.2 equiv) in acetonitrile (15 mL) to which potassium carbonate (0.200 g, 1.45 mmol, 2.9 equiv) was added. After workup and purification with column chromatography (20% EtOAc in hexane as eluent), compound **19d** (0.123 g, 0.38 mmol) was obtained in 76% yield as yellow solids: ^1^H-NMR (acetone-*d_6_*, 500 MHz) δ 5.06 (d, *J* = 17.0 Hz, 1 H, CHCH_2(trans)_), 5.16 (d, *J* = 10.0 Hz, 1 H, CHCH_2(cis)_), 5.41 (d, *J* = 5.0 Hz, 2 H, NCH_2_), 6.07–6.15 (m, 1 H, CHCH_2_), 7.39 (t, *J* = 7.5 Hz, 1 H, ArCH), 7.47 (t, *J* = 7.5 Hz, 1 H, ArCH), 7.52 (d, *J* = 8.0 Hz, 2H, 2 × ArCH), 7.65 (d, *J* = 8.0 Hz, 1H, ArCH), 7.85 (d, *J* = 7.5 Hz, 2H, 2 × ArCH), 7.86 (d, *J* = 8.5 Hz, 1H, ArCH), 7.87 (d, *J* = 16.0 Hz, 1H, COCH), 8.29 (d, *J* = 16.0 Hz, 1H, PhCH); ^13^C-NMR (acetone-*d_6_*, 125 MHz) δ 48.27, 112.42, 117.33, 122.64, 124.61, 124.87, 126.90, 130.16, 131.23, 134.46, 134.70, 136.97, 137.60, 142.93, 143.02, 147.58, 182.99; IR (neat) 3789 (s), 3352 (s), 3187 (m), 2920 (s), 2850 (s), 1663 (C=O, s), 1605 (C=N, s), 1477 (m), 1449 (w), 1402 (w), 1334 (C–N, s), 1300 (m), 1232 (m), 1163 (m), 1056 (s), 988 (s), 919 (m), 830 (s), 742 (s); MS (ESI) *m/z* calcd for C_19_H_15_ClN_2_O: 322.0873, found: 322.0873.

*(2E)-3-**P**henyl-1-(1-(2-(pyrrolidin-1-yl)ethyl)-1H-benzimidazol-2-yl)-2-propen-1-one* (**20a**). Standard procedure **2** was followed by use of compound **18a** (0.124 g, 0.50 mmol, 1.0 equiv) and 1-(2-chloroethyl)pyrrolidine (0.0802 g, 0.60 mmol, 1.2 equiv) in acetonitrile (15 mL) to which potassium carbonate (0.200 g, 1.45 mmol, 2.9 equiv) was added. After workup and purification with column chromatography (75% EtOAc in hexane as eluent), compound **20a** (0.110 g, 0.32 mmol) was obtained in 64% yield as yellow liquid: ^1^H-NMR (acetone-*d_6_*, 500 MHz) δ 1.65 (br s, 4 H, NCH_2_CH_2_), 2.52 (br s, 4 H, NCH_2_CH_2_), 2.84 (t, *J* = 6.5 Hz, 2 H, pyrrolidine-CH_2_CH_2_), 4.83 (t, *J* = 6.5 Hz, 2 H, piperidine-CH_2_CH_2_), 7.37 (t, *J* = 7.5 Hz, 1H, ArCH), 7.46 (t, *J* = 7.5 Hz, 1H, ArCH), 7.50 (d, *J* = 6.5 Hz, 3 H, 3 × ArCH), 7.69 (d, *J* = 8.0 Hz, 1 H, ArCH), 7.83 (d, *J* = 8.0 Hz, 2 H, 2 × ArCH), 7.85 (d, *J* = 8.0 Hz, 1 H, ArCH), 7.90 (d, *J* = 16.0 Hz, 1 H, COCH), 8.24 (d, *J* = 16.0 Hz, 1 H, PhCH); ^13^C-NMR (acetone-*d_6_*, 125 MHz) δ 24.25, 44.80, 54.78, 56.07, 112.24, 122.38, 124.36, 124.41, 126.64, 129.65, 130.00, 131.71, 135.79, 137.55, 142.76, 144.58, 148.35, 183.25; IR (neat) 3650 (s), 3359 (s), 2920 (s), 2850 (s), 2363 (m), 1660 (C=O, s), 1632 (s), 1602 (C=N, s), 1470 (s), 1333 (C–N, s), 1243 (m), 1166 (m), 1040 (s), 745 (s), 697 (m); MS (ESI) *m/z* calcd for C_22_H_23_N_3_O: 345.1841, found: 345.1842**.**

*(**2**E)-1-(1-(2-(**P**yrrolidin-1-yl)ethyl)-1H-benzimidazol-2-yl)-3-(**4**-**methylphen**yl)-2-propen-1-one* (**20b**). Standard procedure **2** was followed by use of compound **18b** (0.131 g, 0.50 mmol, 1.0 equiv) and 1-(2-chloroethyl)pyrrolidine (0.0802 g, 0.60 mmol, 1.2 equiv) in acetonitrile (15 mL) to which potassium carbonate (0.200 g, 1.45 mmol, 2.9 equiv) was added. After workup and purification with column chromatography (60% EtOAc in hexane as eluent), compound **20b** (0.134 g, 0.37 mmol) was obtained in 74% yield as brown oil: ^1^H-NMR (acetone-*d_6_*, 500 MHz) δ 1.65 (p, *J* = 3.5 Hz, 4 H, NCH_2_CH_2_), 2.40 (s, 3 H, PhCH_3_), 2.52 (br s, 4 H, NCH_2_CH_2_), 2.85 (t, *J* = 6.5 Hz, 2 H, pyrrolidine-CH_2_CH_2_), 4.85 (t, *J* = 6.5 Hz, 2 H, pyrrolidine-CH_2_CH_2_), 7.33 (d, *J* = 7.5 Hz, 2 H, 2 × ArCH), 7.37 (t, *J* = 7.5 Hz, 1 H, ArCH), 7.46 (t, *J* = 7.5 Hz, 1 H, ArCH), 7.70 (d, *J* = 8.5 Hz, 1 H, ArCH), 7.73 (d, *J* = 8.0 Hz, 2 H, 2 × ArCH), 7.85 (d, *J* = 8.0 Hz, 1 H, ArCH), 7.88 (d, *J* = 16.0 Hz, 1 H, COCH), 8.23 (d, *J* = 16.0 Hz, 1 H, PhCH); ^13^C-NMR (acetone-*d_6_*, 125 MHz) δ 21.59, 24.39, 45.13, 54.98, 56.45, 112.26, 122.46, 123.63, 124.27, 126.42, 129.74, 130.77, 133.26, 137.73, 142.28, 143.00, 144.55, 148.82, 183.48; IR (neat) 3775 (m), 3357 (m), 2967 (m), 2879 (s), 2785 (m), 2291 (m), 1705 (s), 1660 (C=O, s), 1596 (C=N, s), 1476 (w), 1449 (w), 1407 (w), 1366 (m), 1327 (C–N, m) 1231 (s), 1037 (m), 982 (w), 894 (w), 812 (s), 755 (m), 741 (m), 727 (w); MS (ESI) *m/z* calcd for C_23_H_25_N_3_O: 359.1997, found: 359.1998.

*(**2**E)-3-(4-**M**ethoxyphenyl)-1-(1-(2-(pyrrolidin-1-yl)ethyl)-1H-benzimidazol-2-yl)-2-**prop**en-1-o**n**e* (**20c**). Standard procedure **2** was followed by use of compound **18c** (0.139 g, 0.50 mmol, 1.0 equiv) and 1-(2-chloroethyl)pyrrolidine (0.0802 g, 0.60 mmol, 1.2 equiv) in acetonitrile (15 mL) to which potassium carbonate (0.200 g, 1.45 mmol, 2.9 equiv) was added. After workup and purification with column chromatography (75% EtOAc in hexane as eluent), compound **20c** (0.125 g, 0.33 mmol) was obtained in 66% yield as brown oil: ^1^H-NMR (acetone-*d_6_*, 500 MHz) δ 1.65 (p, *J* = 3.0 Hz, 4 H, NCH_2_CH_2_), 2.53 (br s, 4 H, NCH_2_CH_2_), 2.85 (t, *J* = 6.5 Hz, 2 H, pyrrolidine-CH_2_CH_2_), 3.87 (s, 3 H, OCH_3_), 4.84 (t, *J* = 6.5 Hz, 2 H, pyrrolidine-CH_2_CH_2_), 7.05 (d, *J* = 8.5 Hz, 2 H, 2 × ArCH), 7.36 (t, *J* = 7.5 Hz, 1 H, ArCH), 7.45 (t, *J* = 7.5 Hz, 1 H, ArCH), 7.68 (d, *J* = 8.0 Hz, 1 H, ArCH), 7.79 (d, *J* = 9.0 Hz, 2 H, 2 × ArCH), 7.84 (d, *J* = 9.5 Hz, 1 H, ArCH), 7.87 (d, *J* = 16.0 Hz, 1 H, COCH), 8.10 (d, *J* = 16.0 Hz, 1H, PhCH); ^13^C-NMR (acetone-*d_6_*, 125 MHz) δ 24.36, 45.06, 54.94, 55.94, 56.93, 112.21, 115.52, 122.16, 122.39, 124.23, 126.33, 128.54, 131.57, 137.67, 142.95, 144.48, 148.83, 163.12, 183.38; IR (neat) 3846 (s), 3736 (s), 3650 (m), 3358 (s), 2921 (s), 2851 (s), 2368 (m), 1661 (C=O, s), 1590 (C=N, s), 1511 (s), 1458 (s), 1333 (C–N, s), 1289 (m), 1254 (s), 1171 (s), 1036 (s), 829 (s), 744 (s); MS (ESI) *m/z* calcd for C_23_H_25_N_3_O_2_: 375.1947, found: 375.1948.

*(**2**E)-3-(4-**C**hlorophenyl)-1-(1-(2-(pyrrolidin-1-yl)ethyl)-1H-benzimidazol-2-yl)-2-**prop**en-1-o**n**e* (**20d**). Standard procedure **2** was followed by use of compound **18d** (0.141 g, 0.50 mmol, 1.0 equiv) and 1-(2-chloroethyl)pyrrolidine (0.0802 g, 0.60 mmol, 1.2 equiv) in acetonitrile (15 mL) to which potassium carbonate (0.200 g, 1.45 mmol, 2.9 equiv) was added. After workup and purification with column chromatography (80% EtOAc in hexane as eluent), compound **20d** (0.094 g, 0.25 mmol) was obtained in 50% yield as brown oil: ^1^H-NMR (acetone-*d_6_*, 500 MHz) δ 1.67 (p, *J* = 3.0 Hz, 4 H, NCH_2_CH_2_), 2.56 (br s, 4 H, NCH_2_CH_2_), 2.87 (t, *J* = 6.5 Hz, 2H, pyrrolidine-CH_2_CH_2_), 4.85 (t, *J* = 6.5 Hz, 2 H, pyrrolidine-CH_2_CH_2_), 7.38 (t, *J* = 7.5 Hz, 1 H, ArCH), 7.47 (t, *J* = 8.0 Hz, 1 H, ArCH), 7.54 (d, *J* = 8.0 Hz, 2 H, 2 × ArCH), 7.71 (d, *J* = 8.5 Hz, 1 H, ArCH), 7.85 (d, *J* = 8.5 Hz, 1 H, ArCH), 7.87 (d, *J* = 8.5 Hz, 2 H, 2 × ArCH), 7.88 (d, *J* = 16.0 Hz, 1 H, ketone-CH), 8.23 (d, *J* = 16.0 Hz, 1 H, ph-CH); ^13^C-NMR (acetone-*d_6_*, 125 MHz) δ 24.36, 45.09, 54.96, 56.39, 112.33, 122.49, 124.40, 125.27, 126.60, 130.21, 131.24, 134.81, 136.95, 140.46, 142.92, 183.23; IR (neat) 3352 (s), 2995 (m), 2912 (m), 2307 (m), 1759 (s), 1654 (C=O, s), 1629 (C=N, s), 1465 (m), 1331 (C–N, w), 1246 (s), 1009 (s); MS (ESI) *m/z* calcd for C_22_H_22_ClN_3_O: 379.1451, found: 379.1450**.**

*(**2**E)-3-**P**henyl-1-(1-(2-(piperidin-1-yl)ethyl)-1H-benzimidazol-2-yl)-2-**prop**en-1-one* (**21a**). Standard procedure **2** was followed by use of compound **18a** (0.124 g, 0.50 mmol, 1.0 equiv) and 1-(2-chloroethyl)piperidine (0.0886 g, 0.60 mmol, 1.2 equiv) in acetonitrile (15 mL) to which potassium carbonate (0.200 g, 1.45 mmol, 2.9 equiv) was added. After workup and purification with column chromatography (33% EtOAc in hexane as eluent), compound **21a** (0.120 g, 0.33 mmol) was obtained in 66% yield as yellow liquid: ^1^H-NMR (acetone-*d_6_*, 500 MHz) δ 1.33 (p, *J* = 5.5 Hz, 2 H, NCH_2_CH_2_CH_2_), 1.42 (p, *J* = 5.5 Hz, 4 H, NCH_2_CH_2_CH_2_), 2.40 (br s, 4 H, NCH_2_CH_2_CH_2_), 2.66 (t, *J* = 6.5 Hz, 2 H, piperidine-CH_2_CH_2_), 4.81 (t, *J* = 6.5 Hz, 2 H, piperidine-CH_2_CH_2_), 7.36 (dd, *J* = 7.5 Hz, 1 H, ArCH), 7.45 (dd, *J* = 7.5 Hz, 1 H, ArCH), 7.49 (d, *J* = 7.0 Hz, 3 H, 3 × ArCH), 7.67 (d, *J* = 8.0 Hz, 1 H, ArCH), 7.82 (d, *J* = 8.0 Hz, 2 H, 2 × ArCH), 7.85 (d, *J* = 8.0 Hz, 1 H, ArCH), 7.89 (d, *J* = 16.0 Hz, 1 H, COCH), 8.28 (d, *J* = 16.0 Hz, 1 H, PhCH); ^13^C-NMR (acetone-*d_6_*, 125 MHz) δ 25.03, 26.83, 43.84, 55.67, 59.39, 112.36, 122.44, 124.27, 124.55, 126.42, 129.62, 130.00, 131.65, 135.89, 137.76, 142.87, 144.41, 148.51, 183.32; IR (neat) 3786 (w), 3357 (m), 3055 (w), 3022 (w), 2879 (s), 2775 (w), 2302 (w), 1758 (s), 1662 (C=O, s), 1599 (C=N, s), 1476 (s), 1448 (m), 1407 (m), 1369 (m), 1330 (C–N, s), 1240 (s), 1161 (m), 1117 (w), 1043 (s), 982 (s), 894 (w), 744 (s), 697 (s); MS (ESI) *m/z* calcd for C_23_H_25_N_3_O: 359.1998, found: 359.1996.

*(**2**E)-1-(1-(2-(**P**iperidin-1-yl)ethyl)-1H-benzimidazol-2-yl)-3-(**4-methylphenyl**)-2-**prop**en-1-one* (**21b**). Standard procedure **2** was followed by use of compound **18b** (0.131 g, 0.50 mmol, 1.0 equiv) and 1-(2-chloroethyl)piperidine (0.0886 g, 0.60 mmol, 1.2 equiv) in acetonitrile (15 mL) to which potassium carbonate (0.200 g, 1.45 mmol, 2.9 equiv) was added. After workup and purification with column chromatography (33% EtOAc in hexane as eluent), compound **21b** (0.110 g, 0.29 mmol) was obtained in 58% yield as brown oil: ^1^H-NMR (acetone-*d_6_*, 500 MHz) δ 1.43 (p, *J* = 5.5 Hz, 6 H, NCH_2_CH_2_CH_2_ and NCH_2_CH_2_CH_2_), 2.40 (s, 4 H, NCH_2_CH_2_CH_2_), 2.41 (s, 3 H, ph-CH_3_), 2.67 (t, *J* = 6.5 Hz, 2 H, piperidin-CH_2_CH_2_), 4.84 (t, *J* = 6.5 Hz, 2 H, piperidin-CH_2_CH_2_), 7.32 (d, *J* = 7.5 Hz, 2 H, 2 × ArCH), 7.37 (t, *J* = 7.5 Hz, 1H, ArCH), 7.45 (t, *J* = 7.5 Hz, 1 H, ArCH), 7.69 (d, *J* = 8.5 Hz, 1 H, ArCH), 7.72 (d, *J* = 8.0 Hz, 2 H, 2 × ArCH), 7.84 (d, *J* = 8.0 Hz, 1 H, ArCH), 7.88 (d, *J* = 16.0 Hz, 1 H, COCH), 8.23 (d, *J* = 16.0 Hz, 1H, PhCH); ^13^C-NMR (acetone-*d_6_*, 125 MHz) δ 21.59, 25.07, 26.86, 43.86, 55.72, 59.45, 112.41, 122.43, 123.63, 124.25, 126.36, 129.72, 130.74, 133.26, 137.81, 142.23, 142.94, 144.58, 148.70, 183.47; IR (neat) 3788 (s), 3663 (w), 3574 (w), 3355 (s), 2918 (s), 2846 (s), 1764 (w), 1657 (C=O, w), 1632 (w), 1596 (C=N, w), 1465 (w), 1246 (w), 1325 (C–N, w), 1300 (w), 1246 (s), 988 (w); MS (ESI) *m/z* calcd for C_24_H_27_N_3_O: 373.2154, found: 373.2152.

*(**2**E)-3-(4-**methoxy**phenyl)-1-(1-(2-(piperidin-1-yl)ethyl)-1H-benzimidazol-2-yl)-2-**prop**en-1-one* (**21c**). Standard procedure **2** was followed by use of compound **18c** (0.139 g, 0.50 mmol, 1.0 equiv) and 1-(2-chloroethyl)piperidine (0.0886 g, 0.60 mmol, 1.2 equiv) in acetonitrile (15 mL) to which potassium carbonate (0.200 g, 1.45 mmol, 2.9 equiv) was added. After workup and purification with column chromatography (40% EtOAc in hexane as eluent), compound **21c** (0.166 g, 0.43 mmol) was obtained in 86% yield as brown oil: ^1^H-NMR (acetone-*d_6_*, 500 MHz) δ 1.33 (p, *J* = 5.5 Hz, 2 H, NCH_2_CH_2_CH_2_), 1.42 (p, *J* = 5.5 Hz, 4 H, NCH_2_CH_2_CH_2_), 2.41 (s, 4 H, NCH_2_CH_2_CH_2_), 2.67 (t, *J* = 6.5 Hz, 2 H, piperidin-CH_2_CH_2_), 3.89 (s, 3 H, OCH_3_), 4.84 (t, *J* = 6.5 Hz, 2 H, piperidin-CH_2_CH_2_), 7.06 (d, *J* = 8.5 Hz, 2 H, 2 × ArCH), 7.36 (t, *J* = 7.5 Hz, 1 H, ArCH), 7.45 (t, *J* = 7.5 Hz, 1 H, ArCH), 7.69 (d, *J* = 8.0 Hz, 1 H, ArCH), 7.80 (d, *J* = 9.0 Hz, 2 H, 2 × ArCH), 7.84 (d, *J* = 8.0 Hz, 1 H, ArCH), 7.87 (d, *J* = 16.0 Hz, 1 H, ketone-CH), 8.14 (d, *J* = 16.0 Hz, 1 H, ph-CH); ^13^C-NMR (acetone-*d_6_*, 125 MHz) δ 25.09, 26.89, 43.87, 55.74, 55.95, 112.40, 115.55, 122.24, 122.38, 124.19, 126.26, 128.59, 131.56, 137.81, 142.96, 144.47, 148.85, 163.14, 183.45; IR (neat) 3780 (s), 3357 (s), 3187 (s), 2921 (s), 2851 (s), 1661 (C=O, s), 1630 (s), 1591 (C=N, s), 1569 (m), 1512 (s), 1473 (m), 1331 (C–N, s), 1255 (s), 1172 (s), 1040 (s), 829 (m), 740 (m); MS (ESI) *m/z* calcd for C_24_H_27_N_3_O_2_: 389.2103, found: 389.2103.

*(**2**E)-3-(4-**C**hlorophenyl)-1-(1-(2-(piperidin-1-yl)ethyl)-1H-benzimidazol-2-yl)-2-**prop**en-1-one* (**21d**). Standard procedure **2** was followed by use of compound **18d** (0.141 g, 0.50 mmol, 1.0 equiv) and 1-(2-chloroethyl)piperidine (0.0886 g, 0.60 mmol, 1.2 equiv) in acetonitrile (15 mL) to which potassium carbonate (0.200 g, 1.45 mmol, 2.9 equiv) was added. After workup and purification with column chromatography (25% EtOAc in hexane as eluent), compound **21d** (0.098 g, 0.25 mmol) was obtained in 50% yield as yellow solids: ^1^H-NMR (acetone-*d_6_*, 500 MHz) δ 1.33 (p, *J* = 5.5 Hz, 2 H, NCH_2_CH_2_CH_2_), 1.42 (p, *J* = 5.5 Hz, 4 H, NCH_2_CH_2_CH_2_), 2.41 (s, 4 H, NCH_2_CH_2_CH_2_), 2.67 (t, *J* = 6.5 Hz, 2 H, piperidin-CH_2_CH_2_), 4.85 (t, *J* = 6.5 Hz, 2 H, piperidin-CH_2_CH_2_), 7.38 (t, *J* = 7.5 Hz, 1 H, ArCH), 7.47 (t, *J* = 7.5 Hz, 1 H, ArCH), 7.55 (d, *J* = 8.5 Hz, 2 H, 2 × ArCH), 7.71 (d, *J* = 8.5 Hz, 1 H, ArCH), 7.85 (d, *J* = 8.5 Hz, 1 H, ArCH), 7.88 (d, *J* = 9.0 Hz, 2 H, 2 × ArCH), 7.88 (d, *J* = 16.0 Hz, 1 H, COCH), 8.27 (d, *J* = 16.0 Hz, 1 H, PhCH); ^13^C-NMR (acetone-*d_6_*, 125 MHz) δ 25.06, 26.87, 43.89, 55.72, 59.44, 112.49, 122.48, 124.36, 125.33, 126.54, 130.20, 131.22, 134.83, 136.93, 137.85, 142.92, 148.55, 183.28; IR (neat) 3840 (s), 3736 (s), 3650 (m), 3568 (m), 3363 (s), 2921 (s), 2851 (s), 2362 (s), 1666 (C=O, s), 1604 (C=N, s), 1490 (s), 1405 (s), 1334 (C–N, s), 1300 (m), 1164 (s), 1093 (s), 1043 (s), 894 (w), 822 (s), 741 (s), 497 (m); MS (ESI) *m/z* calcd for C_23_H_24_ClN_3_O: 393.1608, found: 393.1609.

*(**2**E)-1-(1-(2-**M**orpholinoethyl)-1H-benzimidazol-2-yl)-3-phenyl-2-**prop**en-1-one* (**22a**). Standard procedure **2** was followed by use of compound **18a** (0.124 g, 0.50 mmol, 1.0 equiv) and 4-(2-chloroethyl)morpholine (0.0898 g, 0.60 mmol, 1.2 equiv) in acetonitrile (15 mL) was added potassium carbonate (0.200 g, 1.45 mmol, 2.9 equiv). After workup and purification with column chromatography (33% EtOAc in hexane as eluent), compound **22a** (0.152 g, 0.42 mmol) was obtained in 84% yield as yellow solids: ^1^H-NMR (acetone-*d_6_*, 500 MHz) δ 2.46 (t, *J* = 5.0 Hz, 4 H, NCH_2_CH_2_O), 2.74 (t, *J* = 6.5 Hz, 2 H, morpholine-CH_2_CH_2_), 3.50 (t, *J* = 5.0 Hz, 4 H, OCH_2_), 4.88 (t, *J* = 6.5 Hz, 2 H, morpholine-CH_2_CH_2_), 7.38 (t, *J* = 7.5 Hz, 1 H, ArCH), 7.47 (t, *J* = 7.5 Hz, 1 H, ArCH), 7.51 (d, *J* = 7.0 Hz, 3 H, 3 × ArCH), 7.72 (d, *J* = 8.0 Hz, 1 H, ArCH), 7.85 (d, *J* = 7.0 Hz, 2 H, 2 × ArCH), 7.86 (d, *J* = 7.0 Hz, 1 H, ArCH), 7.91 (d, *J* = 16.0 Hz, 1 H, COCH), 8.29 (d, *J* = 16.0 Hz, 1 H, PhCH); ^13^C-NMR (acetone-*d_6_*, 125 MHz) δ 43.43, 54.86, 59.12, 67.47, 112.41, 122.47, 124.41, 124.55, 126.57, 129.71, 130.08, 131.79, 135.88, 137.71, 142.86, 144.72, 148.59, 183.52; IR (neat) 3786 (s), 3352 (s), 2921 (s), 2851 (s), 2308 (w), 1665 (C=O, s), 1602 (C=N, s), 1446 (br), 1410 (m), 1333 (C–N, s), 1300 (m), 1116 (s), 1067 (m), 1040 (m), 867 (w), 744 (s), 554 (w); MS (ESI) *m/z* calcd for C_22_H_23_N_3_O_2_: 361.1790, found: 361.1792.

*(**2**E)-1-(1-(2-**M**orpholinoethyl)-1H-benzimidazol-2-yl)-3-(**4-methylphen**yl)-2-**prop**en-1-one* (**22b**). Standard procedure **2** was followed by use of compound **18b** (0.131 g, 0.50 mmol, 1.0 equiv) and 4-(2-chloroethyl)morpholine (0.0898 g, 0.60 mmol, 1.2 equiv) in acetonitrile (15 mL) to which potassium carbonate (0.200 g, 1.45 mmol, 2.9 equiv) was added. After workup and purification with column chromatography (33% EtOAc in hexane as eluent), compound **22b** (0.119 g, 0.32 mmol) was obtained in 64% yield as brown oil: ^1^H-NMR (acetone-*d_6_*, 500 MHz) δ 2.39 (s, 3 H, PhCH_3_), 2.45 (t, *J* = 5.0 Hz, 4 H, NCH_2_CH_2_O), 2.72 (t, *J* = 6.5 Hz, 2 H, morpholine-CH_2_CH_2_), 3.50 (t, *J* = 5.0 Hz, 4 H, OCH_2_), 4.86 (t, *J* = 6.5 Hz, 2 H, morpholine-CH_2_CH_2_), 7.32 (d, *J* = 8.0 Hz, 2 H, 2 × ArCH), 7.37 (t, *J* = 7.5 Hz, 1 H, ArCH), 7.46 (t, *J* = 7.5 Hz, 1 H, ArCH), 7.70 (d, *J* = 8.0 Hz, 1 H, ArCH), 7.72 (d, *J* = 8.0 Hz, 2 H, 2 × ArCH), 7.85 (d, *J* = 7.0 Hz, 1 H, ArCH), 7.87 (d, *J* = 16.0 Hz, 1 H, COCH), 8.23 (d, *J* = 16.0 Hz, 1H, PhCH); ^13^C-NMR (acetone-*d_6_*, 125 MHz) δ 21.59, 43.42, 54.87, 59.15, 67.48, 112.35, 122.46, 123.58, 124.32, 126.45, 129.75, 130.74, 133.19, 137.71, 142.30, 142.90, 144.72, 148.66, 183.54; IR (neat) 3786 (s), 3357 (s), 3187 (m), 2920 (s), 2846 (s), 1660 (C=O, s), 1632 (m), 1597 (C=N, s), 1470 (br), 1410 (m), 1333 (C–N, s), 1303 (m), 1180 (m), 1114 (s), 985 (m), 867 (m), 815 (m), 744 (s); MS (ESI) *m/z* calcd for C_23_H_25_N_3_O_2_: 375.1947, found: 375.1948.

*(**2**E)-3-(4-**M**ethoxyphenyl)-1-(1-(2-morpholinoethyl)-1H-benzimidazol-2-yl)-2-**prop**en-1-one* (**22c**). Standard procedure **2** was followed by use of compound **18c** (0.139 g, 0.50 mmol, 1.0 equiv) and 4-(2-chloroethyl)morpholine (0.0898 g, 0.60 mmol, 1.2 equiv) in acetonitrile (15 mL) to which potassium carbonate (0.200 g, 1.45 mmol, 2.9 equiv) was added. After workup and purification with column chromatography (33% EtOAc in hexane as eluent), compound **22c** (0.122 g, 0.31 mmol) was obtained in 62% yield as yellow oil: ^1^H-NMR (acetone-*d_6_*, 500 MHz) δ 2.45 (t, *J* = 5.0 Hz, 4 H, NCH_2_CH_2_O), 2.72 (t, *J* = 6.5 Hz, 2 H, morpholine-CH_2_CH_2_), 3.50 (t, *J* = 5.0 Hz, 4 H, OCH_2_), 3.88 (s, 3 H, OCH_3_), 4.86 (t, *J* = 6.5 Hz, 2 H, morpholine-CH_2_CH_2_), 7.05 (d, *J* = 8.0 Hz, 2 H, 2 × ArCH), 7.37 (t, *J* = 7.5 Hz, 1H, ArCH), 7.45 (t, *J* = 7.5 Hz, 1 H, ArCH), 7.68 (d, *J* = 8.5 Hz, 1 H, ArCH), 7.79 (d, *J* = 8.5 Hz, 2 H, 2 × ArCH), 7.85 (d, *J* = 8.5 Hz, 1 H, ArCH), 7.87 (d, *J* = 16.0 Hz, 1 H, COCH), 8.15 (d, *J* = 16.0 Hz, 1H, PhCH); ^13^C-NMR (acetone-*d_6_*, 125 MHz) δ 43.40, 43.86, 55.93, 59.14, 67.47, 112.28, 115.50, 122.15, 122.40, 124.25, 126.33, 128.49, 131.57, 137.67, 142.88, 144.60, 148.75, 163.11, 183.54; IR (neat) 3789 (s), 3663 (w), 3359 (br), 2922 (s), 2851 (s), 1645 (C=O, s), 1591 (C=N,s), 1512 (w), 1468 (br), 1333 (C–N, m), 1289 (m), 1256 (s), 1173 (s), 1114 (s), 1067 (m), 1040 (m), 829 (w), 741 (w); MS (ESI) *m/*z calcd for C_23_H_25_N_3_O_3_: 391.1896, found: 391.1897.

*(**2**E)-3-(4-**C**hlorophenyl)-1-(1-(2-morpholinoethyl)-1H-benzimidazol-2-yl)-2-**prop**en-1-one* (**22d**). Standard procedure **2** was followed by use of compound **18d** (0.141 g, 0.50 mmol, 1.0 equiv) and 4-(2-chloroethyl)morpholine (0.0898 g, 0.60 mmol, 1.2 equiv) in acetonitrile (15 mL) to which potassium carbonate (0.200 g, 1.45 mmol, 2.9 equiv) was added. After workup and purification with column chromatography (25% EtOAc in hexane as eluent), compound **22d** (0.123 g, 0.31 mmol) was obtained in 62% yield as yellow solids: ^1^H-NMR (acetone-*d_6_*, 500 MHz) δ 2.45 (t, *J* = 5.0 Hz, 4 H, NCH_2_CH_2_O), 2.73 (t, *J* = 6.5 Hz, 2 H, morpholine-CH_2_CH_2_), 3.49 (t, *J* = 5.0 Hz, 4 H, OCH_2_), 4.88 (t, *J* = 6.5 Hz, 2 H, morpholine-CH_2_CH_2_), 7.38 (t, *J* = 7.5 Hz, 1 H, ArCH), 7.47 (t, *J* = 7.5 Hz, 1 H, ArCH), 7.54 (d, *J* = 8.5 Hz, 2 H, 2 × ArCH), 7.72 (d, *J* = 8.0 Hz, 1 H, ArCH), 7.85 (d, *J* = 8.0 Hz, 1 H, ArCH), 7.88 (d, *J* = 8.5 Hz, 2 H, 2 × ArCH), 7.88 (d, *J* = 16.0 Hz, 1 H, COCH), 8.28 (d, *J* = 16.0 Hz, 1 H, PhCH); ^13^C-NMR (acetone-*d_6_*, 125 MHz) δ 43.48, 54.91, 59.17, 67.51, 112.44, 122.54, 124.42, 125.31, 126.62, 130.21, 131.25, 134.79, 136.98, 137.80, 142.95, 143.04, 148.53, 183.36; IR (neat) 3357 (s), 3192 (s), 2920 (s), 2850 (s), 1662 (C=O, s), 1632 (s), 1602 (C=N, s), 1564 (m), 1489 (m), 1470 (s), 1410 (s), 1333 (C–N, s) 1300(s), 1114 (s), 1034 (s), 1010 (s), 823 (s), 740 (s); MS (ESI) *m/z* calcd for C_22_H_22_ClN_3_O_2_: 395.1401, found: 395.1403.

*(**2**E)-1-(1-(3-**M**orpholinopropyl)-1H-benzimidazol-2-yl)-3-phenyl-2-**prop**en-1-one* (**23a**). Standard procedure **2** was followed by use of compound **18a** (0.124 g, 0.50 mmol, 1.0 equiv) and 4-(3-chloropropyl)morpholine (0.0982 g, 0.60 mmol, 1.2 equiv) in acetonitrile (15 mL) to which potassium carbonate (0.200 g, 1.45 mmol, 2.9 equiv) was added. After workup and purification with column chromatography (66.7% EtOAc in hexane as eluent), compound **23a** (0.116 g, 0.31 mmol) was obtained in 62% yield as yellow oil: ^1^H-NMR (acetone-*d_6_*, 500 MHz) δ 2.30 (s, 6 H, NCH_2_CH_2_O and morpholine-CH_2_CH_2_CH_2_), 2.33 (m, 8 H, morpholine-CH_2_CH_2_CH_2_), 3.57 (t, *J* = 4.5 Hz, 4 H, OCH_2_), 4.79 (t, *J* = 7.0 Hz, 2 H, morpholine-CH_2_CH_2_CH_2_), 7.37 (t, *J* = 7.5 Hz, 1 H, ArCH), 7.45 (t, *J* = 7.5 Hz, 1 H, ArCH), 7.49 (d, *J* = 7.0 Hz, 3 H, 3 × ArCH), 7.75 (d, *J* = 8.0 Hz, 1 H, ArCH), 7.81 (d, *J* = 8.0 Hz, 1 H, ArCH), 7.82 (d, *J* = 8.0 Hz, 1 H, ArCH), 7.86 (d, *J* = 7.0 Hz, 1 H, ArCH), 7.88 (d, *J* = 16.0 Hz, 1 H, COCH), 8.33 (d, *J* = 16.0 Hz, 1 H, PhCH); ^13^C-NMR (acetone-*d_6_*, 125 MHz) δ 27.45, 44.26, 54.32, 56.24, 67.27, 112.39, 122.38, 124.24, 124.39, 126.58, 129.64, 129.96, 131.68, 135.75, 137.78, 142.64, 144.48, 147.85, 183.09; IR (neat) 3780 (s), 3687 (s), 2918 (w), 2846 (w), 2808 (w), 1770 (m), 1663 (C=O, s), 1599 (C=N, s), 1476 (m), 1457 (w), 1407 (m), 1333 (C–N, s), 1306 (m), 1117 (s), 1043 (m), 979 (m), 859 (m), 744 (s), 686 (m); MS (ESI) *m/z* calcd for C_23_H_25_N_3_O_2_: 375.1947, found: 375.1944.

*(**2**E)-1-(1-(3-**M**orpholinopropyl)-1H-benzimidazol-2-yl)-3-(**4-methylphen**yl)-2-**prop**en-1-one* (**23b**). Standard procedure **2** was followed by use of compound **18b** (0.131 g, 0.50 mmol, 1.0 equiv) and 4-(3-chloropropyl)morpholine (0.0982 g, 0.60 mmol, 1.2 equiv) in acetonitrile (15 mL) to which potassium carbonate (0.200 g, 1.45 mmol, 2.9 equiv) was added. After workup and purification with column chromatography (40% EtOAc in hexane as eluent), compound **23b** (0.108 g, 0.28 mmol) was obtained in 56% yield as brown oil: ^1^H-NMR (acetone-*d_6_*, 500 MHz) δ 2.30 (m, 6 H, NCH_2_CH_2_O and morpholine-CH_2_CH_2_CH_2_), 2.35 (t, *J* = 6.5 Hz, 2 H, morpholine-CH_2_CH_2_CH_2_), 2.39 (s, 3 H, ph-CH_3_), 3.57 (t, *J* = 4.5 Hz, 4 H, OCH_2_), 4.81 (t, *J* = 7.5 Hz, 2 H, morpholine-CH_2_CH_2_CH_2_), 7.32 (d, *J* = 8.0 Hz, 2 H, 2 × ArCH), 7.37 (t, *J* = 7.0 Hz, 1 H, ArCH), 7.46 (t, *J* = 7.5 Hz, 1 H, ArCH), 7.70 (d, *J* = 8.0 Hz, 2 H, 2 × ArCH), 7.76 (d, *J* = 8.0 Hz, 1 H, ArCH), 7.86 (d, *J* = 8.0 Hz, 1 H, ArCH), 7.86 (d, *J* = 16.0 Hz, 1 H, COCH), 8.28 (d, *J* = 16.0 Hz, 1 H, PhCH); ^13^C-NMR (acetone-*d_6_*, 125 MHz) δ 21.59, 27.57, 44.28, 54.47, 56.33, 67.40, 112.45, 122.46, 123.41, 124.33, 126.48, 129.74, 130.72, 133.20, 137.94, 142.26, 142.84, 144.50, 148.12, 183.26; IR (neat) 3840 (s), 3736 (s), 3673 (m), 3650 (m), 3568 (m), 2921 (s), 2851 (s), 2348 (s), 1663 (C=O, s), 1600 (C=N, s), 1454 (w), 1407 (w), 1333 (C–N, s), 1180 (s), 1117 (s), 1036 (s), 985 (w), 814 (s), 742 (m); MS (ESI) *m/z* calcd for C_24_H_27_N_3_O_2_: 389.2103, found: 389.2101.

*(**2**E)-3-(4-**M**ethoxyphenyl)-1-(1-(3-morpholinopropyl)-1H-benzimidazol-2-yl)-2-**prop**en-1-one* (**23c**). Standard procedure **2** was followed by use of compound **18c** (0.139 g, 0.50 mmol, 1.0 equiv) and 4-(3-chloropropyl)morpholine (0.0982 g, 0.60 mmol, 1.2 equiv) in acetonitrile (15 mL) to which potassium carbonate (0.200 g, 1.45 mmol, 2.9 equiv) was added. After workup and purification with column chromatography (50% EtOAc in hexane as eluent), compound **23c** (0.174 g, 0.43 mmol) was obtained in 86% yield as yellow oil: ^1^H-NMR (acetone-*d_6_*, 500 MHz) δ 2.31–2.34 (m, 8 H, NCH_2_CH_2_O, morpholine-CH_2_CH_2_CH_2_ and morpholine-CH_2_CH_2_CH_2_), 3.57 (t, *J* = 4.5 Hz, 4 H, OCH_2_), 3.87(s, 3 H, OCH_3_), 4.80 (t, *J* = 7.5 Hz, 2 H, morpholine-CH_2_CH_2_CH_2_), 7.04 (d, *J* = 8.5 Hz, 2 H, 2 × ArCH), 7.36 (t, *J* = 7.5 Hz, 1 H, ArCH), 7.44 (t, *J* = 7.5 Hz, 1 H, ArCH), 7.74 (d, *J* = 8.0 Hz, 1 H, ArCH), 7.78 (d, *J* = 9.0 Hz, 1 H, ArCH), 7.84 (d, *J* = 8.0 Hz, 1 H, ArCH), 7.85 (d, *J* = 16.0 Hz, 1 H, COCH), 8.18 (d, *J* = 16.0 Hz, 1 H, PhCH); ^13^C-NMR (acetone-*d_6_*, 125 MHz) δ 29.62, 44.26, 54.50, 55.94, 56.34, 67.43, 112.41, 115.51, 122.03, 122.43, 124.24, 126.34, 128.54, 131.57, 137.96, 142.89, 144.36, 148.27, 163.11, 183.21; IR (neat) 3359 (s), 3186 (s), 2921 (s), 2851 (s), 1660 (C=O, s), 1632 (s), 1591 (C=N,s), 1511 (s), 1471 (s), 1418 (w), 1410 (w), 1331 (C–N, s), 1255 (s), 1172 (s), 1116 (s), 1036 (s), 828 (s), 741 (s); MS (ESI) *m/z* calcd for C_24_H_27_N_3_O_3_: 405.2052, found: 405.2051.

*(**2**E)-3-(4-**C**hlorophenyl)-1-(1-(3-morpholinopropyl)-1H-benzimidazol-2-yl)-2-**prop**en-1-one* (**23d**). Standard procedure **2** was followed by use of compound **18d** (0.141 g, 0.50 mmol, 1.0 equiv) and 4-(3-chloropropyl)morpholine (0.0982 g, 0.60 mmol, 1.2 equiv) in acetonitrile (15 mL) to which potassium carbonate (0.200 g, 1.45 mmol, 2.9 equiv) was added. After workup and purification with column chromatography (33% EtOAc in hexane as eluent), compound **23d** (0.115 g, 0.28 mmol) was obtained in 56% yield as brown oil: ^1^H-NMR (acetone-*d_6_*, 500 MHz) δ 2.32 (br s, 4 H, NCH_2_CH_2_O), 2.36 (t, *J* = 6.5 Hz, 2 H, morpholine-CH_2_CH_2_CH_2_), 2.67 (s, 2 H, morpholine-CH_2_CH_2_CH_2_), 3.57 (t, *J* = 4.5 Hz, 4 H, OCH_2_), 4.80 (t, *J* = 7.0 Hz, 2H, morpholine- CH_2_CH_2_CH_2_), 7.38 (t, *J* = 7.5 Hz, 1 H, ArCH), 7.46 (t, *J* = 7.5 Hz, 1H, ArCH), 7.54 (d, *J* = 8.5 Hz, 2H, 2 × ArCH), 7.78 (d, *J* = 8.0 Hz, 1H, ArCH), 7.86 (d, *J* = 8.0 Hz, 1H, ArCH), 7.87 (d, *J* = 8.5 Hz, 2H, 2 × ArCH), 7.88 (d, *J* = 16.0 Hz, 1H, ketone-CH), 8.32 (d, *J* = 16.0 Hz, 1H, ph-CH); ^13^C-NMR (acetone-*d_6_*, 125 MHz) δ 27.56, 44.33, 54.50, 56.35, 67.42, 112.55, 122.55, 124.44, 125.16, 126.66, 128.94, 129.61, 130.20, 131.24, 134.81, 138.03, 142.84, 148.01, 183.11; IR (neat) 3788 (s), 3663 (w), 3573 (w), 3359 (m), 2920 (s), 2850 (s), 1665 (C=O, s), 1602 (C=N, s), 1407 (w), 1334 (C–N, s), 1300 (w), 1117 (s), 1092 (s), 1043 (s), 821 (s), 742 (s); MS (ESI) *m/z* calcd for C_23_H_24_ClN_3_O_2_: 409.1557, found: 409.1557.

### 3.4. MTT Assay

Assays to measure proliferation, viability, and cytotoxicity are commonly used to monitor the response and health of cells in culture after treatment with various stimuli. First, cell-lines were seeded in 96 well plates with 3000–5000 cells/well. On the second day, cell-lines were treated with different compounds in various concentrations of 100, 50, 25, 12.5, 6.25, 0 µg/µL. After 48 h, the MTT reagent 5(6)-carboxyfluorescein diacetate *N*-succinimidyl ester (CFSE) (5 mg/mL) was diluted to one tenth in culture medium as the MTT assay solution. After the addition of MTT assay solution in 96-well plates (100 µL/well), the sample was incubated at 37 °C for 1–1.5 h. Cell counting using viability dyes such as trypan blue or calcein-AM can provide both the rate of proliferation as well as the percentage of viable cells. CFSE is a popular choice for measuring the number of cellular divisions a population has undergone. Upon entering the cell, CFSE is cleaved by intracellular esterases to form the fluorescent compound and the succinimidyl ester group covalently reacts with primary amines on intracellular proteins. After labeling the cells with MTT, as described above, all but 100 µL of medium was removed from the wells. Then DMSO (50 µL) was added to each well and mixed thoroughly with a pipette and incubated at 37 °C for 10 min. After incubation, the sample was gently shaken and absorbance was read at 560 nm by an ELISA reader.

### 3.5. Flow Cytometry

Cell cycle analysis of compounds **20a**–**23a** was performed by propidium iodide (PI) staining followed by flow cytometry in OVCAR-3 cells. OVCAR-3 cells were seeded into 6-cm dishes at a density of 2 × 10^5^ cells/dish for 24 h. Cells were then treated with serial dilution of compound **20a**–**23a** for 48 h. The cells were harvested and fixed with ice cold 70% ethanol at 4 °C for 24 h. After the washes, cells were stained with PI staining solution (20 mg/mL PI in 0.3 mL of PBS containing 200–400 unit of RNase A) followed by incubation at 37 °C for 30 min in the dark. The cells were analyzed by flow cytometry (Accuri C6, Becton Dickenson, Bergen, NJ, USA) to determine the proportion of cells within the cycle.

## 4. Conclusions

In conclusion, we describe the synthesis and examination of a new series of N-substituted benzimidazole derivatives with a functional chalcone group. On the basis of the MTT assay against A549, MCF-7, HEP-G2 and OVCAR-3 cells, we confirmed that these derivatives exhibit considerable potential as anticancer drugs. In particular, compound **23a** ((2E)-1-(1-(3-morpholinopropyl)-1H-benzimidazol-2-yl)-3-phenyl-2-propen-1-one) attained high IC_50_ values on A549, MCF-7, HEP-G2 and OVCAR-3 cells, and showed in vitro cytotoxicity comparable or superior to that of cisplatin. Among the tested derivatives, 11 compounds (**20a, 20c, 20d, 21a**–**21c, 22c, 22d, 23a**–**23c**) had IC_50_ values between 10.34 and 14.88 μM, which were lower than 16.04 μM (the IC_50_ value of cisplatin), for the OVCAR-3 cells. According to the MTT results, the nitrogen-containing 5- or 6-membered ring in the N-substituted benzimidazole derivatives can lead to enhanced cytotoxic effects on MCF-7 and OVCAR-3 cells, which might serve as new templates in the synthesis and development of potent therapeutics.

Additionally, the flow cytometry results indicate OVCAR-3 cells treated with compound **20a** and **21a** were 55% and 63% in the G2/M phase, respectively (Table 2). The activities of compounds **20a** and **21a** may be related to their abilities to control cells entering apoptosis from the S or G2/M phase of the cell cycle. A similar mechanism was also found in thyroid carcinoma, BHT-101, cells after treatment with 50 μM curcumin [57]. Moreover, compound **23a** had low IC_50_ concentration in OVCR-3 cells, but it did not show any significant change in flow cytometry results. Cytotoxicity of compound **23a** may have occurred through cell necrosis.

To further improve the efficacy and specificity for cancer cell death, we conclude that the novel series of N-substituted benzimidazole derivatives can serve as prototype molecules for further development of a new class of anti-cancer agents.

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
