# Peer review of "Design and Synthesis of Benzimidazole-Chalcone Derivatives as Potential Anticancer Agents"

_molecules, 2019, doi:10.3390/molecules24183259_

Round 1

Reviewer 1 Report

Title: Design and Synthesis of Benzimidazole-Chalcone Derivatives as Potential Anticancer Agents

The manuscript describes the synthesis and anticancer properties of a series of benzimidazole-chalcone derivatives.

The paper is now suitable for publication. 

Author Response

Reviewer 1

The manuscript describes the synthesis and anticancer properties of a series of benzimidazole-chalcone derivatives.

The paper is now suitable for publication.

Response 1:

Thank you very much for your comments.  We have revised the manuscript carefully and asked a native english speaker, Ms. Miranda Loney, to revise this manuscript. We believe that the language is now acceptable for publish.

Reviewer 2 Report

Line 103, 123, and 242: The concentration notation of hydrochloric acid is incorrect. It is 4.0 M instead of 4.0 N.

The following words were recommended to rewrite as follows.
Line 223: “magnesium sulfate” is “anhydrous magnesium sulfate.”
Line 256, 266, and 274: “MgSO4(s)” is “anhydrous MgSO4.”

Since the NMR specification has already been shown, there are deleted from the compound assignments.

The assignment of OH signal is not found in compound 16.

The NMR assignments of compound 17 dose not match the chart in Supplementary material.

It is recommended to rewrite the text of “Standard procedure 1…” as follows.
A 40% aqueous solution of potassium hydroxide (2.0 mL, 2.0 mmol) was added to a stirred solution of 2-acetylbenzimidazole 17 (0.640 g, 4.0 mmol) and substituted aromatic aldehydes (4.4 mmol) in ethanol (8 mL). The mixture was stirred at room temperature for 10 h. After the consumption of the compound 17, the reaction mixture was quenched with water (2-4 mL) and extracted with ethyl acetate (10-20 mL). The combined organic layers were washed with brine (5.0 mL), dried over anhydrous magnesium sulfate, filtered, and concentrated under reduced pressure. The residue was purified by use of column chromatography (15-20% EtOAc in hexane as eluent) to give the desired benzimidazolyl-chalcone derivatives 18a-d.
(2E)-1-(1H-benzimidazol-2-yl)-3-phenyl-2-propen-1-one (18a). 0.870 g (3.51 mmol, 88% yield); 15% EtOAc in hexane; yellow solid; 1H NMR (acetone-d6) δ 7.37 (d, J=8.0 Hz, 1H, ArH), 7.44 (t, J=7.5 Hz, 1H, ArH), 7.52-7.53 (m, 3H, ArH), 7.68 (d, J=8.0 Hz, 1H, ArH), 7.86-7.89 (m, 3H, ArH), 8.03 (d, J=16.0 Hz, 1H, COCH), 8.17 (d, J=16.0 Hz, 1H, CHPh); 13C NMR (acetone-d6) δ 113.64, 122.45, 122.55, 124.18, 126.82, 128.97, 129.78, 130.08, 131.77, 131.91, 135.81, 144.70, 145.23, 146.01, 150.20, 181.92; IR (neat) cm-1 3357 (s), 3247 (N-H, s), 2920 (s), 2850 (s), 1661 (C=O, s), 1632 (m), 1597 (C=N, s), 1424 (s), 1331 (C-N, s), 1215 (m), 1138 (w), 1089 (w), 971 (w), 741 (w), 722 (w); MS (ESI) m/z calcd for C16H12N2O: 248.0950, found: 248.0950. Its spectroscopic characteristics are consistent with those of the same compound reported [56].

It is recommended to rewrite the text of “Standard procedure 2…” as well.

Author Response

Reviewer 2

Point 1: Line 103, 123, and 242: The concentration notation of hydrochloric acid is incorrect. It is 4.0 M instead of 4.0 N.

Response 1: Thank you for your kindly suggestion, we had corrected all the concentration notation according your suggestion.  

Point 2: The following words were recommended to rewrite as follows.

Line 223: “magnesium sulfate” is “anhydrous magnesium sulfate.”

Line 256, 266, and 274: “MgSO4(s)” is “anhydrous MgSO4.”

Response 2: Thank you for your suggestion, we had corrected the manuscript according your suggestion. The “magnesium sulfate” had been corrected to “anhydrous magnesium sulfate” and the “MgSO4(s)” had been corrected to “anhydrous MgSO4.”   

Point 3: Since the NMR specification has already been shown, there are deleted from the compound assignments.

Response 3: Thank you for your comments. In this manuscript 24 synthesized compounds only 4 compounds are reported, we think we have obligation to report the detail of the new synthesized compounds. 

Point 4: The assignment of OH signal is not found in compound 16. 

Response 4: Thank you for your comments.  We are sure about the compound is correct, not only this compound is reported but we have every characterized data.  The OH signal missing was probably due to the moisture in the CDCl3

Point 5: The NMR assignments of compound 17 do not match the chart in Supplementary material.

Response 5: Thank you very much for point out our mistake. After we carefully recheck our laboratory record book, the chemical shift of -COCH3 shall be 2.81. It’s our human error to mark the wrong peak in the supplementary material spectrum. 

Point 6: It is recommended to rewrite the text of “Standard procedure 1…” as follows.

A 40% aqueous solution of potassium hydroxide (2.0 mL, 2.0 mmol) was added to a stirred solution of 2-acetylbenzimidazole 17 (0.640 g, 4.0 mmol) and substituted aromatic aldehydes (4.4 mmol) in ethanol (8 mL). The mixture was stirred at room temperature for 10 h. After the consumption of the compound 17, the reaction mixture was quenched with water (2-4 mL) and extracted with ethyl acetate (10-20 mL). The combined organic layers were washed with brine (5.0 mL), dried over anhydrous magnesium sulfate, filtered, and concentrated under reduced pressure. The residue was purified by use of column chromatography (15-20% EtOAc in hexane as eluent) to give the desired benzimidazolyl-chalcone derivatives 18a-d.

(2E)-1-(1H-benzimidazol-2-yl)-3-phenyl-2-propen-1-one (18a). 0.870 g (3.51 mmol, 88% yield); 15% EtOAc in hexane; yellow solid; 1H NMR (acetone-d6) δ 7.37 (d, J=8.0 Hz, 1H, ArH), 7.44 (t, J=7.5 Hz, 1H, ArH), 7.52-7.53 (m, 3H, ArH), 7.68 (d, J=8.0 Hz, 1H, ArH), 7.86-7.89 (m, 3H, ArH), 8.03 (d, J=16.0 Hz, 1H, COCH), 8.17 (d, J=16.0 Hz, 1H, CHPh); 13C NMR (acetone-d6) δ 113.64, 122.45, 122.55, 124.18, 126.82, 128.97, 129.78, 130.08, 131.77, 131.91, 135.81, 144.70, 145.23, 146.01, 150.20, 181.92; IR (neat) cm-1 3357 (s), 3247 (N-H, s), 2920 (s), 2850 (s), 1661 (C=O, s), 1632 (m), 1597 (C=N, s), 1424 (s), 1331 (C-N, s), 1215 (m), 1138 (w), 1089 (w), 971 (w), 741 (w), 722 (w); MS (ESI) m/z calcd for C16H12N2O: 248.0950, found: 248.0950. Its spectroscopic characteristics are consistent with those of the same compound reported [56].

It is recommended to rewrite the text of “Standard procedure 2…” as well.

Response 6: Thank you very much for your valuable comments that are very helpful for us to improve the quality of our paper.  Now we rewrite the standard procedure 1 and 2 according your suggestion.

Author Response

Reviewer 3

The manuscript is well designed and written, although biological results are only moderate. Manuscript is acceptable for publication after minor revision:

Point 1: Replace “under an atmosphere of nitrogen” with “under nitrogen atmosphere”

Response 1: Thank you for your suggestion.  We had corrected this manuscript according your suggestion, the “under an atmosphere of nitrogen” had been correct to “under nitrogen atmosphere”. 

Point 2: Replace “18a-18d” with “18a-d”, “19a-19d” with “19a-d” etc in the whole manuscript and Scheme 1.

Response 2: Thank you for your kindly suggestion.  We had corrected all the “18a-18d” to “18a-d”, “19a-19d” with “19a-d” etc. The compounds number shown in the Scheme 1 had also been corrected at the same way.

Point 3: NMRs: replace “d6” with “d6

Response 3: Thank you for your suggestion. We had corrected the manuscript according to your suggestion, now every d6 of d-solvent is italic.

Point 4: Conclusions: 23a has low IC50s, not high.

Response 4: Thank you for your correction. We apologize for this mistake. Now the word had been corrected to correctly describe about the IC50 value.

Point 5: The whole manuscript: replace “4 N” with SI unit “4 M”

The whole manuscript: replace “1H” with “1H”all “E” with “E

Lines 116-119: replace all “Allyl” with “allyl”

Response 5: Thank you for your correction. We had corrected the manuscript according to your suggestion, all these error styles/formats had been corrected.

Point 6: The whole manuscript: stop marks should be after, not before reference: replace “, [1]” with “[1].” etc.

Response 6: Thank you for your suggestion. We had corrected the manuscript according to your suggestion, now all the stop mark had been relocated after each citations.